# Economic evaluation of sterilization reversal in infertility treatment: A systematic review

Brandon Chongthanadon[1], Suvijak Untaaveesup[1,2], Chayanis Kositamongkol[3]*, Pochamana Phisalprapa[3], Krasean Panyakhamlerd[4], Vitaya Titapant[5]

1 Faculty of Medicine Siriraj Hospital, Mahidol University, Bangkok, Thailand, 2 Phaholpolpayuhasena Hospital, Kanchanaburi, Thailand, 3 Division of Ambulatory Medicine, Department of Medicine, Faculty of Medicine Siriraj Hospital, Mahidol University, Bangkok, Thailand, 4 Department of Obstetrics and Gynecology, Faculty of Medicine, Chulalongkorn University, Bangkok, Thailand, 5 Division of Maternal-Fetal Medicine, Department of Obstetrics and Gynecology, Faculty of Medicine, Siriraj Hospital, Mahidol University, Bangkok, Thailand

☯ These authors contributed equally to this work.
* chayanis.kos@mahidol.ac.th

## Abstract

### Objectives

Although sterilization is intended to be permanent, some individuals later seek fertility. In such cases, options can be limited and financially burdensome. This review evaluated the cost-effectiveness of sterilization reversal surgery in previously sterilized individuals.

### Methods

We searched MEDLINE, Embase, and Scopus from inception in 1946 through December 2025, following the Preferred Reporting Items for Systematic Reviews and Meta-Analyses guidance. We included studies that analyzed cost-effectiveness or reported the costs of sterilization reversal in males (vasectomy reversal) or females (tubal anastomosis), with assisted reproductive technologies as comparators. Study quality was assessed using the Consolidated Health Economic Evaluation Reporting Standards 2022 checklist. Two authors independently screened each study to reduce bias. All costs per outcome were converted to 2024 United States dollars for analysis and comparisons.

### Results

Of 1628 identified articles, 24 studies met the eligibility criteria. Almost all examined populations in high-income countries, such as the United States, the Netherlands, and Singapore. Thirteen studies evaluated tubal anastomosis, and eleven evaluated vasectomy reversal. Most studies reported lower total costs for sterilization reversal than for assisted reproductive technologies, with comparable outcomes.

**Data availability statement:** All relevant data are within the manuscript and its Supporting Information files.

**Funding:** This research was funded by the Royal Thai College of Obstetricians and Gynaecologists (grant number 011 RTCOG2309). The funders had no role in study design, data collection and analysis, decision to publish, or preparation of the manuscript.

**Competing interests:** The authors have declared that no competing interests exist.

Vasectomy reversal was preferred for male patients irrespective of the female partner's age, whereas tubal anastomosis was preferred for female patients aged 40 years or younger. For older patients, assisted reproductive technologies were more cost-effective.

## Conclusions

Tubal anastomosis and vasectomy reversal may be economically advantageous compared with assisted reproductive technologies for infertility due to prior sterilization. However, societal factors, including a country's socioeconomic context and policy feasibility, should be considered.

## Introduction

Surgical sterilization is a highly effective permanent contraceptive option, particularly attractive to individuals with a well-defined family plan. With no observed pregnancies within 6 months after vasectomy and a 2.9% pregnancy rate within 12 months after tubal sterilization, these sterilization techniques offer a reliable, long-term solution that often surpasses other methods in effectiveness and convenience [1,2].

Sterilization is classified into 2 categories: vasectomy in males and tubal sterilization in females. Vasectomy disrupts sperm transport from the proximal to the distal vas deferens, whereas tubal sterilization occludes the fallopian tube to prevent fertilization [3,4]. However, following the 2026 ESGO consensus statement, opportunistic salpingectomy has increasingly become the standard of care to significantly reduce the risk of tubo-ovarian carcinoma [5]. For individuals who initially opted for vasectomy or tubal sterilization, they may choose surgery or assisted reproductive technologies (ARTs) to regain fertility. On the other hand, if female individuals underwent salpingectomy, ARTs are the only options, as the procedure is anatomically impossible to be reversed. Surgical procedures to regain fertility include vasectomy reversal (VR) for males or tubal anastomosis (TA) for females. ARTs involve manipulation of oocytes or embryos and include in vitro fertilization (IVF), intracytoplasmic sperm injection (ICSI), and related procedures such as cryopreservation and donation of oocytes and embryos [6].

VR, introduced more than a century ago to restore fertility, is performed using 2 techniques [7]. Vasovasostomy anastomoses the severed ends of the vas deferens, whereas vasoepididymostomy (epididymovasostomy) joins the vas deferens to the epididymis. These procedures achieve pregnancy rates of 22%–68% (mean 49%) [8]. TA—also called tubal ligation reversal or tubal re-anastomosis—reconnects the ligated fallopian tube and has a pregnancy rate of 42%–69% [9]. In comparison, ART such as IVF has a pregnancy rate of approximately 46% [10]. However, TA is associated with a higher risk of ectopic pregnancy (6.7%) compared with both IVF (5.6%) and the general population (2%) [11].

Sterilization can lead to regret when life circumstances change and the desire to conceive returns. A retrospective analysis of National Survey of Family Growth data

reported that approximately 10% of females regretted sterilization [12]. Without appropriate management, secondary infertility can impose a burden; therefore, addressing infertility in this population is important.

Cost-effectiveness is a key consideration when choosing between reversal surgery and ARTs. Multiple studies have compared sterilization reversal with ARTs; several reported favorable economic outcomes for reversal, whereas others favored ARTs. Reviews commonly concluded that VR was more cost-effective than ARTs [13,14], but findings for TA were variable [9]. However, comprehensive syntheses remain scarce, and many are narrative reviews without formal search methods, which can yield biased or misleading conclusions. The most recent systematic review reporting the cost-effectiveness of VR, conducted in 2002, emphasized ARTs rather than surgical reversal procedures [15].

To address this gap, this systematic review aimed to evaluate the cost-effectiveness of VR and TA compared with ARTs. Furthermore, by assessing studies without publication date restrictions, this review aims to evaluate the longitudinal shifts in cost-effectiveness driven by technological advancements in both surgeries and ARTs. We anticipate that the findings will inform evidence-based policy planning in infertility care.

## Materials and methods

### Literature search

We conducted a systematic literature search from inception date of each database through December 2025 in accordance with the Preferred Reporting Items for Systematic Reviews and Meta-Analyses protocol [16] (S1 Table). We searched MEDLINE (via PubMed, 1946–2025), Embase (1947–2025), and Scopus (1960–2025) following a protocol registered on PROSPERO (CRD420250588564) [17]. We also performed manual searches of reference lists of included studies and of relevant systematic reviews, narrative reviews, and meta-analyses. The detailed search strategy is provided in S2 Table.

### Inclusion criteria

We included economic evaluations (e.g., cost-effectiveness analyses, cost-utility analyses) of VR in males and TA in females, irrespective of language. The screening of non-English records was facilitated by a digital translation tool (i.e., Google Translate, Google LLC). We also considered clinical studies that reported economic outcomes as a prespecified outcome or secondary endpoint.

Eligible populations comprised individuals seeking to restore fertility after prior sterilization, regardless of sex or demographic characteristics. Populations could include patient cohorts from medical records or simulated cohorts in model-based analyses.

Interventions of interest were VR in males and TA in females, including vasovasostomy and vasoepididymostomy (epididymovasostomy) for VR and tubal ligation reversal or tubal re-anastomosis for TA. Comparators were ARTs, namely IVF, ICSI, and related procedures. We included studies comparing sterilization reversal with ARTs, regardless of terminology used.

### Exclusion criteria

We excluded studies of nonhuman subjects, as well as studies comparing variations of the intervention itself (e.g., 1-layer versus 2-layer VR or manual versus robotic TA). Additionally, studies focusing on same-sex couples were excluded. The primary objective of sterilization reversal is to restore fertility for unassisted reproduction. However, reproduction in same-sex couples inherently requires ARTs or third-party gametes; therefore, evaluating sterilization reversal against ARTs is not clinically applicable in this cohort. We also excluded studies without full text or abstract, systematic or umbrella reviews, and studies using comparators that do not involve manipulation of oocytes or embryos, such as intrauterine insemination.

### Screening process

Two authors (B.C. and S.U.) independently screened titles and abstracts in Covidence, a web-based screening platform. Eligibility questions were addressed during screening; unresolved disagreements prompted full-text review and adjudication by a third author (C.K.). Only studies meeting all inclusion criteria proceeded to data extraction.

### Data extraction

One author (B.C.) extracted data, which a second author (S.U.) then cross-checked. Clinical variables included population demographics (country, sex, age); specified interventions; time horizon; and outcomes, including pregnancy, delivery, or live birth rates. Economic variables included intervention costs, discount rate, analytic perspective, cost per outcome, and the incremental cost-effectiveness ratio. When data were not specified, we recorded "not reported" in the extraction form. We recorded "not applicable" for fields that did not apply to a specific study design (e.g., time horizons in model analyses). We extracted each study's conclusion on whether the intervention was more cost-effective than the comparator and categorized it as "yes," "no," or "inconclusive."

### Data analysis

We categorized included studies into 2 groups based on biological sex: male and female. To compare costs across years, we adjusted each cost per outcome for inflation using the country's healthcare consumer price index (CPI), standardizing to 2024 prices in the local currency. If a study did not report the reference year for currency conversion, we assumed the publication year. We then converted costs to 2024 United States dollars (US dollars, $) using the annual foreign exchange rate [18]. We reported results in a comparison table, categorized by participant sex and sorted by study year. We also stratified data by maternal age because this factor influences infertility [19]. For studies that did not report cost per outcome, i.e., those reporting only incremental costs, or only costs and outcomes separately, we included them in the analysis table as "not reported." For studies reporting multiple interventional pathways, calculation was limited to isolated sterilization reversal arms and isolated ART arms to prevent confounding. Within each intervention category (VR or TA), the option with the lower inflation-adjusted cost per outcome was deemed more cost-effective.

### Study assessment

We assessed study quality using the Consolidated Health Economic Evaluation Reporting Standards (CHEERS) 2022 checklist, a structured framework for reporting economic evaluations [20]. Although developed as a reporting guideline, CHEERS is widely used to appraise health economic evaluations and was suitable for this review [21]. The checklist comprises 28 items spanning clinical and economic dimensions. We rated each item as fulfilled, not fulfilled, or not applicable. Studies meeting more CHEERS criteria were deemed to be of higher quality.

## Results

### Study selection

From the 3 databases, 1615 articles were identified. An additional 13 articles were found through citation searches, yielding 1628 in total. After removing 204 duplicates, 1424 titles and abstracts were screened. During screening, 1080 articles were excluded, leaving 344 articles for full-text retrieval. Of these, 24 were not retrieved due to lack of both full text and abstract, and 296 articles were excluded for the following reasons: 190 were excluded due to irrelevant outcomes; these were studies that did not report pregnancy, delivery, or live birth rates or any economic outcome; 17 were excluded due to ineligible comparators (i.e., studies that did not include ART as the comparator); 16 were excluded due to ineligible interventions; and 73 were excluded due to ineligible study designs, including non-economic studies and those with ineligible populations. After full evaluation, 24 articles were included for data extraction and outcome interpretation (Fig 1).

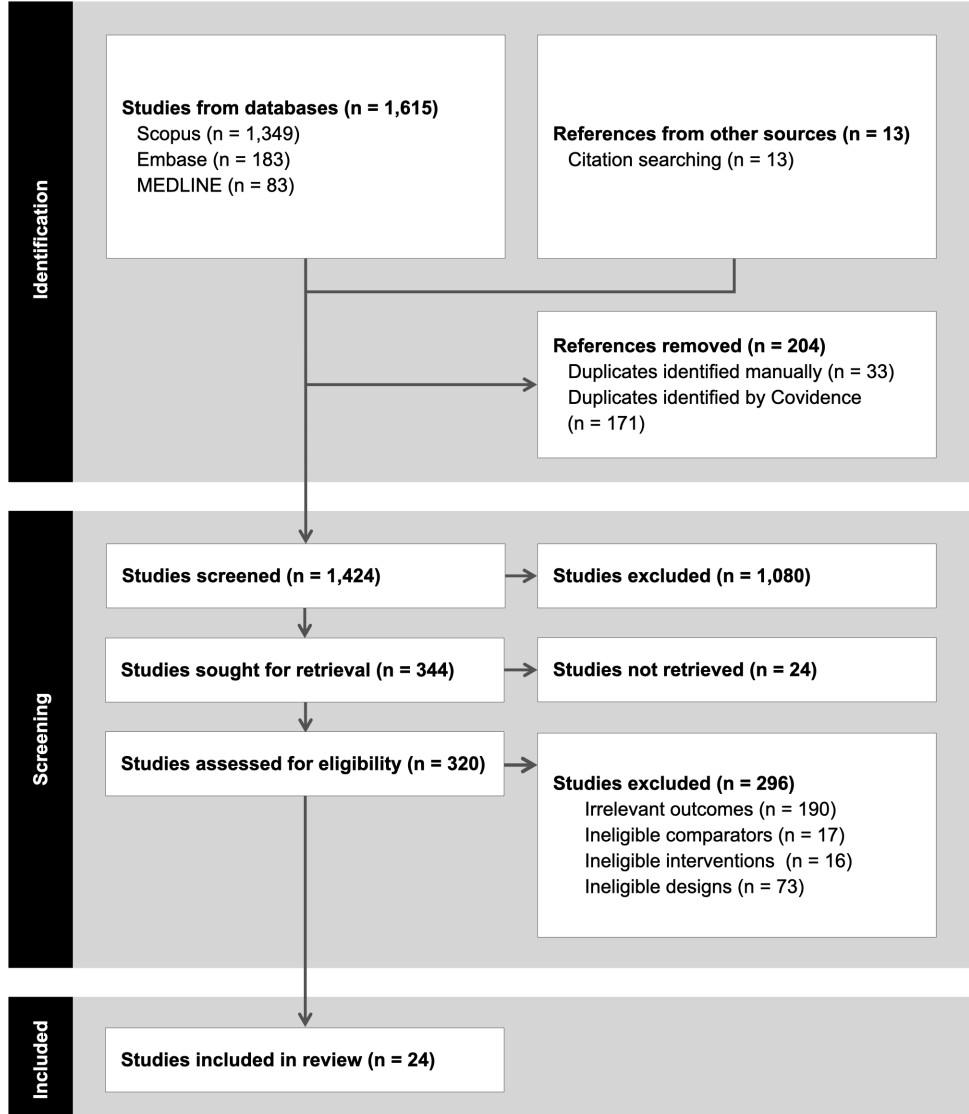

**Fig 1. PRISMA diagram.**

## Study characteristics

Most studies (23 of 24) were conducted in high-income countries, with 16 from the United States. Two each were from the Netherlands [22,23] and Singapore [24,25], and 1 each from Australia [26], Belgium [27], Germany [28]. The remaining study [29] was conducted in China, the sole upper-middle-income country represented. Seven studies [22,23,30–34] were published between 1990–1999, two studies [28,35] were published between 2000–2002. Of these, only five studies [30,32–35] were included in the previous systematic review [15]; the remaining 15 studies were conducted after 2002.

## Study designs and evaluation types

Cost-effectiveness analyses predominated (*n* = 21), with 2 cost-utility analyses [36,37] and 1 cost-minimization analysis [29]. Model-based analyses were most common, accounting for 11 studies, followed by retrospective (*n* = 10) and

prospective cohort (*n* = 3), respectively. Thirteen studies evaluated TA in females, and 11 evaluated VR in males. Most were 2-arm studies (*n* = 17), with fewer 3-arm (*n* = 4) and 4-arm (*n* = 3) designs (**Fig 2**).

## Time horizons and perspectives

Time horizons were reported primarily for cohort-based cost-effectiveness and cost-utility analyses. Nearly all cost-effectiveness analyses [22,24–28,30–33,35] and 1 cost-minimization analysis cohort study [29] reported time horizons of 2–11 years; 1 study did not report a horizon [23]. However, time horizons were marked as "not applicable" for static model-based analyses, as these specific studies evaluate costs and outcomes at a single point in time rather than tracking a cohort longitudinally. For studies' perspective, only 6 of the 23 studies reported the analytic perspective [24,32,37–40]. Four used the patient perspective [24,32,37,38], 1 used the healthcare provider perspective [39], and 1 used the societal perspective [40]. Detailed characteristics are presented in **Table 1**.

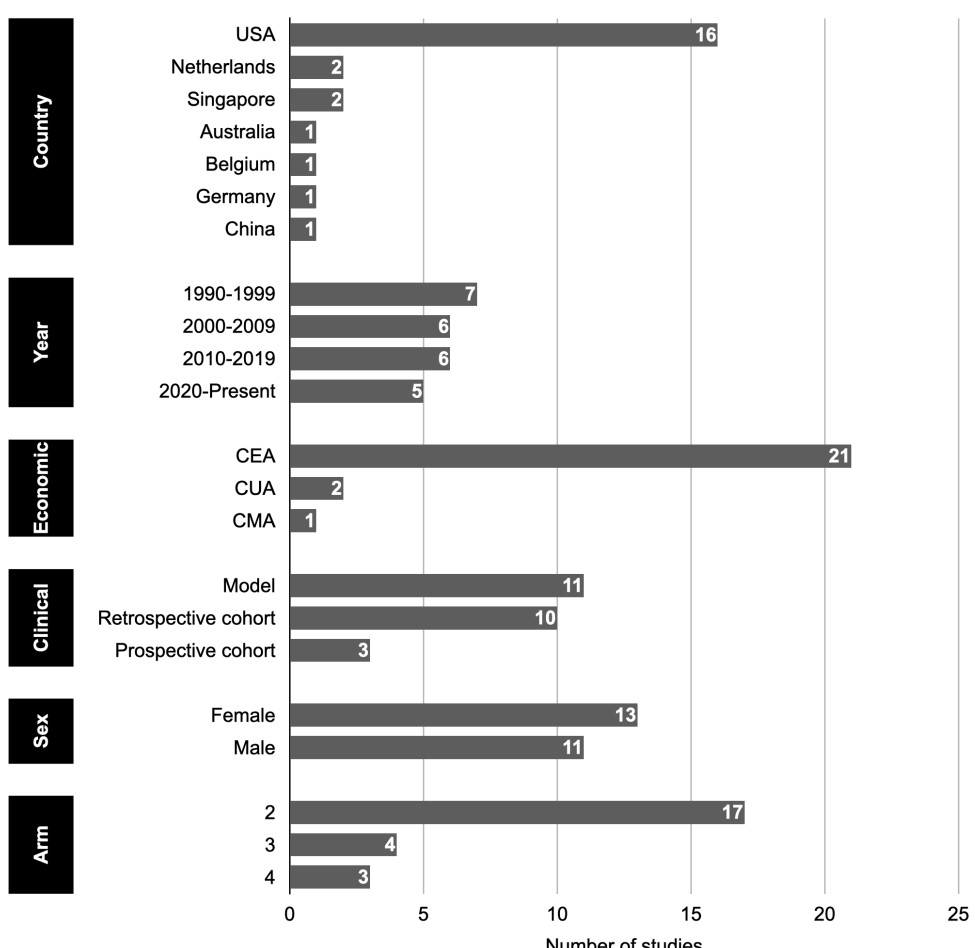

**Fig 2. Characteristics of included studies.**

**Table 1. Characteristics of included studies.**

| Study (Authors, year) | Country | Design | Population | Maternal age (years) | Perspective | Time horizon | Intervention (number of patients) | Arm |
|---|---|---|---|---|---|---|---|---|
| **Tubal anastomosis** | | | | | | | | |
| Alford et al, 2010* [38] | USA | CEA, model | Hypothetical patients | NR | Patient | NA | I: Bilateral TA | 2 |
| | | | | | | | C: IVF | |
| Boeckxstaens et al, 2007 [27] | Belgium | CEA, retrospective cohort | Patients from Dutch speaking Brussels Free University, Belgium | All age | NR | 1990-2005 (6 years) | I: TA ($n = 84$) | 2 |
| | | | | | | | C: IVF ($n = 79$) | |
| Chua et al, 2020 [25] | Singapore | CEA, retrospective cohort | Previously ligated female patients | <40 | NR | 2011-2016 (6 years) | I: TA ($n = 12$) | 2 |
| | | | | | | | C: IVF ($n = 31$) | |
| Copperman et al, 1996 [30] | USA | CEA, prospective cohort | Female patients who underwent tubal surgery and has on concomitant male factor infertility. | <40 | NR | 1993-1994 (2 years) | I: TA ($n = 31$) | 2 |
| | | | | | | | C: IVF ($n = 67$) | |
| Haan et al, 1991 [22] | Netherlands | CEA, prospective cohort | Patients underwent IVF in 5 Dutch hospitals | All age | NR | 1986-1988 (3 years) | I: TA | 2 |
| | | | | | | | C: Three cycles of IVF ($n = 1462$ for all IVF) | |
| Haan et al, 1992 [23] | Netherlands | CEA, prospective cohort | Patients underwent IVF in 5 Dutch hospitals | All age | NR | NR | I: TA | 2 |
| | | | | | | | C: IVF | |
| Hirshfeld-Cytron et al, 2013 [39] | USA | CEA, model | Hypothetical patients | All age | Healthcare provider | NA | I1: TA with more favorable prognosis (clip or ring tubal ligation) | 3 |
| | | | | | | | I2: TA with less favorable prognosis (postpartum tubal ligation, electrocautery, or unknown type) | |
| | | | | | | | I3: IVF | |
| Holst et al, 1991 [31] | USA | CEA, retrospective cohort | Female patients who underwent TA versus IVF | All age | NR | 1980-1989 (10 years) | I1: TA in 1980–1982 ($n = 206$) | 4 |
| | | | | | | | I2: TA in 1986–1988 ($n = 104$) | |
| | | | | | | | C1: All IVF in 1986–1989 ($n = 389$) | |
| | | | | | | | C2: Completed IVF treatment in 1986–1989 ($n = 128$) | |
| Messinger et al, 2015 [40] | USA | CEA, model | Hypothetical patients | All age | Society | NA | I: TA | 2 |
| | | | | | | | C: IVF | |
| Petrucco et al, 2007 [26] | Australia | CEA, retrospective cohort | Female patients who underwent microsurgical TA | ≥40 | NR | 1997-2005 (9 years) | I1: microsurgical TA in Adelaide, South Australia ($n = 35$) | 3 |
| | | | | | | | I2: microsurgical TA in Missouri, USA ($n = 12$) | |
| | | | | | | | C: IVF | |
| Tan et al, 2010 [24] | Singapore | CEA, retrospective cohort | Female patients who underwent TA without any known semen abnormalities | <40 | Patient | 1998-2008 (11 years) | I1: laparoscopic TA ($n = 9$) | 3 |
| | | | | | | | I2: open TA (n = 10) | |
| | | | | | | | C: IVF ($n = 327$) | |
| Winter et al, 2012* [41] | USA | CEA, model | Hypothetical patients | All age | NR | NA | I1: TA with more favorable prognosis (clip or ring tubal ligation) | 3 |
| | | | | | | | I2: TA with less favorable prognosis (postpartum tubal ligation, electrocautery, or unknown type) | |
| | | | | | | | I3: IVF | |

*(Continued)*

**Table 1.** (Continued)

| Study (Authors, year) | Country | Design | Population | Maternal age (years) | Perspective | Time horizon | Intervention (number of patients) | Arm |
|---|---|---|---|---|---|---|---|---|
| Womack et at., 2020* [42] | USA | CEA, model | Hypothetical patients | NR | NR | NA | I: TA | 2 |
| | | | | | | | C: IVF | |
| **Vasectomy reversal** | | | | | | | | |
| Cheng et al, 2021 [37] | USA | CUA, model | Hypothetical male patients with prior vasectomy | ≥35 | Patient | NA | I1: VR, then NC | 4 |
| | | | | | | | I2: SR, then IVF/ICSI | |
| | | | | | | | I3: back-up VR/SR, then IVF/ICSI, followed by NC | |
| | | | | | | | I4: back-up VR/SR, then NC, followed by IVF/ICSI | |
| Craig et al, 2017* [36] | USA | CUA, model | Hypothetical male patients with prior vasectomy | All age | NR | NA | I1: VR followed by NC | 4 |
| | | | | | | | I2: TESE followed by IVF | |
| | | | | | | | I3: back-up VR, then NC, followed by IVF | |
| | | | | | | | I4: back-up VR, then IVF, followed by NC | |
| Deck et al, 2000 [35] | USA | CEA, retrospective cohort | Male patients underwent microsurgical vasectomy reversal with ovulating female partner | >37 | NR | 1994-1998 (5 years) | I: Microsurgical VR (n=29) | 2 |
| | | | | | | | C: TESE and IVF/ICSI | |
| Donovan et al, 1998 [32] | USA | CEA, retrospective cohort | Male patients with prior failed microscopic VR | All age | Patient | 1986-1996 (11 years) | I: Repeat VR (n=18) | 2 |
| | | | | | | | C: MESA and ICSI/IVF (n=9) | |
| Heidenreich et al, 2000 [28] | Germany | CEA, retrospective cohort | Male patients who underwent microsurgical double-layer VVS for VR | All age | NR | 1993-1998 (6 years) | I: VVS (n=112) | 2 |
| | | | | | | | C: MESA/TESE and ICSI (n=69) | |
| Kassab et al, 2024* [43] | USA | CEA, model | Hypothetical male patients with prior vasectomy | 25-45 | NR | NA | I: VR | 2 |
| | | | | | | | C: IVF | |
| Kolettis et al, 1997 [33] | USA | CEA, retrospective cohort | Male patients who underwent VES | All age | NR | 1979-1995 (7 years) | I: VES (n=55) | 2 |
| | | | | | | | C: MESA and ICSI | |
| Lee et al, 2008 [44] | USA | CEA, model | Hypothetical infertile male patients | All age | NR | NA | I: VR | 2 |
| | | | | | | | C: TESE/MESA and IVF | |
| Meng et al, 2005 [45] | USA | CEA, model | Hypothetical infertile male patients with varicocele or post-vasectomy obstruction | All age | NR | NA | I: VR | 2 |
| | | | | | | | C: SR and ICSI | |
| Pavlovich et al, 1997 [34] | USA | CEA, model | Hypothetical male patients with prior vasectomy | ≤39 | NR | NA | I: Microsurgical VR | 2 |
| | | | | | | | C: SR and ICSI | |
| Zhang et al, 2021 [29] | China | CMA, retrospective cohort | Male patients with obstructive azoospermia | <40 | NR | 2018-2019 (2 years) | I: microsurgical VES (n=65) | 2 |
| | | | | | | | C: TESE/PESA with ICSI (n=341) | |

* Only abstracts were available.

**Abbreviations:** C, comparator; CEA, cost-effectiveness analysis; CMA, cost-minimization analysis; CUA, cost-utility analysis; I, intervention; ICSI, intracytoplasmic sperm injection; IVF, in vitro fertilization; MESA, microscopic epididymal sperm aspiration; n, number; NA, not applicable; NC, natural conception; NR, not reported; PESA, percutaneous epididymal sperm aspiration; SR, sperm retrieval; TA, tubal anastomosis; TESE, testicular sperm extraction; VES, vasoepididymostomy; VR, vasectomy reversal; VVS, vasovasostomy.

### Study populations

**TA studies (female sterilization reversal).** Eight studies used patient records, with or without age restrictions [22–27,30,31]. Three reported results for females younger than 40 years [24,25,30]. Only 1 study reported results for females aged 40 years or older [26]. Three additional TA studies used hypothetical patients without age restrictions; all were model-based analyses [39–41]. Two other model-based analyses in the TA group did not specify population details [38,42].

**VR studies (male sterilization reversal).** Almost half of the VR studies (*n* = 5) included real patients [28,29,32,33,35]. Among these, 1 study specified female partners older than 37 years [35], and 1 specified female partners younger than 40 years [29]. The remaining 6 studies modeled hypothetical male patients [34,36,37,43–45]. None reported male partner age; however, 1 specified female partners aged 35 years or older [37], 1 specified female partners aged 39 years or younger [34], and one stratified female partners aged from 25 to 45 years. Three studies specified the surgical technique: vasoepididymostomy [29,33] or vasovasostomy [28].

### Clinical and economic outcomes

**TA outcomes.** Across TA studies [22–27,30,31,38–42], outcomes were reported as pregnancy, delivery, cumulative delivery, or live birth rates. As expected, pregnancy rates exceeded delivery and live birth rates, but values varied widely. Pregnancy rates for TA ranged from 5.0% [40] to 77.8% [24], compared with 10% [23,40] to 46.8% [24] for ARTs. Delivery rates in 1 study ranged from 36.6% to 72.2% for TA and from 51.4% to 52.4% for ARTs [27]. Live birth rates ranged from 23.7% [31] to 97.7% [39] for TA and from 12% [38] to 75.6% [39] for ARTs.

Almost all TA studies reported costs; however, 2 did not specify cost details [41,42]. Only 5 studies adjusted costs over time within their study periods [26,38–41].

**VR outcomes.** VR studies reported pregnancy, delivery, and live birth rates; additionally, 2 reported quality-adjusted life years [36,37]. Eight VR studies reported total costs, either with a cost breakdown [28,32–34] or without detailed cost information [29,35,37,43]. Two model-based studies reported procedural costs only [44,45], and 1 did not specify cost details [36]. Only 1 study adjusted costs over time using a medical care CPI [44].

Details are summarized in **Table 2**.

### Cost-effectiveness findings

**Cost per outcome reporting.** In TA studies, cost-effectiveness was reported by clinical outcome; 2 studies reported incremental cost per live birth rather than cost per live birth [39,41]. In the VR group [28,29,32–37,43–45], nearly all studies reported cost per outcome; however, 1 cost-minimization analysis reported only intervention costs [29], and 1 cost-effective analysis reported only total costs [43].

**TA findings.** In total, 9 of 13 studies concluded that TA was more cost-effective than ART. One study favored TA regardless of patient age or other characteristics [38]. However, 7 studies favored TA over ARTs for females below specific ages: below 37 years in 1 study [27], below 40 years in 3 studies [24,25,42], and below 41 years in 3 studies [39–41]. One study favored TA for females older than 40 years [26]. Two studies found ART to be as cost-effective as TA [22,23], and 2 favored ART over TA [30,31].

**VR findings.** All VR studies concluded that VR was more cost-effective than ARTs for males seeking to restore fertility after prior vasectomy. Detailed cost-effectiveness data are presented in **Table 3**.

### Inflation-adjusted analyses

**Adjustment methods.** Using the reported cost per outcome, we inflation-adjusted the costs from 17 studies to 2024 local currency values. We applied the Australian healthcare CPI [46], Belgian healthcare expenditure CPI [47], German

**Table 2. Interventions, outcomes, and costs.**

| Study (Authors, year) | Intervention | Source of outcomes | Outcomes | Source of costs | Conversion year | Discount rate | Costs |
|---|---|---|---|---|---|---|---|
| **Tubal anastomosis** | | | | | | | |
| Alford et al, 2010 [38] | | Literature review | *Live birth rates by maternal age* | Literature review | 2012 | NR | *Total Cost* |
| | I: Bilateral TA | | I: (<35) 47%, (41) 36% | | | | I: $6551-$9593 |
| | C: IVF | | C: (<35) 41%, (41) 12% | | | | C: $10 000-$21 000 |
| Boeckxstaens et al, 2007 [27] | | Institutional records | *Cumulative delivery rates (by maternal age)* | Literature review and institutional records | NR | NR | *Total cost* |
| | I: TA | | I: average 59.5%, (<37) 72.2%, (≥37) 36.6% | | | | I: medical cost of €2360, nonmedical cost of €1448 |
| | C: IVF | | C: average 52.0%, (<37) 52.4%, (≥37) 51.4% | | | | C: medical cost of €2426, nonmedical cost of €575 |
| Chua et al, 2020 [25] | | Institutional records | *Pregnancy rates / live birth rates* | Institutional records | NR | NR | *Total cost* |
| | I: TA | | I: 75.0% / 58.3% | | | | I: SG$15 132 |
| | C: IVF | | C: 35.5% / 25.8% | | | | C: Frozen embryo transfer SG$4000, fresh cycle SG$12 500, 31 fresh and 8 thaw cycles were performed |
| Copperman et al, 1996 [30] | | Institutional records | *Pregnancy rates* | Institutional records | NR | NR | *Total cost* |
| | I: TA | | I: 19.3% | | | | I: $4696 |
| | C: IVF | | C: 25.0% | | | | C: $5830 |
| Haan et al, 1991 [22] | | Institutional records | *Pregnancy rate* | Institutional records | NR | NR | *Total cost* |
| | I: TA | | I: 30% | | | | I: NLG 5500-NLG 7700 |
| | C: Three cycles of IVF | | C: 30% | | | | C: NLG 2500 per cycle |
| Haan et al, 1992 [23] | | Institutional records | *Pregnancy rate* | Institutional records | NR | NR | *Total cost* |
| | I: TA | | I: 30% | | | | I: NLG 5500-NLG 7700 |
| | C: IVF | | C: 10% | | | | C: NLG 2500 per cycle |
| Hirshfeld-Cytron et al, 2013 [39] | | Literature review | *Live birth rates (by maternal age)* | Literature review | 2012 | CPI | *Total cost (by maternal age)* |
| | I1: TA with more favorable prognosis | | I1: (≤35) 97.7%, (35–40) 93.7%, (>40) 76.8% | | | | I1: (≤35) $19 300, (35–40) $23 000, (>40) $26 200 |
| | I2: TA with less favorable prognosis | | I2: (≤35) 95.3%, (35–40) 91.9%, (>40) 74.3% | | | | I2: (≤35) $20 700, (35 –40) $25 000, (>40) $27 200 |
| | I3: IVF | | I3: (≤35) 75.4%, (35–40) 75.6%, (>40) 48.5% | | | | I3: (≤35) $26 200, (35 –40) $31 000, (>40) $28 900 |

*(Continued)*

| Study (Authors, year) | Intervention | Source of outcomes | Outcomes | Source of costs | Conversion year | Discount rate | Costs |
|---|---|---|---|---|---|---|---|
| Holst et al, 1991 [31] | | Institutional records | *Live birth rates* | Institutional records | NR | NR | *Total cost* |
| | I1: TA in 1980–1982 | | I: 23.7% | | | | I: $4000 |
| | I2: TA in 1986–1988 | | | | | | |
| | C1: All IVF in 1986–1989 | | C: 72.3% | | | | C: $1500 per cycle |
| | C2: Completed IVF treatment in 1986–1989 | | | | | | |
| Messinger et al, 2015 [40] | | Literature review | *Ongoing pregnancy rates (by maternal age)* | Literature review | 2014 | CPI | *Total cost* |
| | I: TA | | I: (<35) 63%, (35–40) 44%, (>40) 5% | | | | I: $8685 |
| | C: IVF | | C: (<35) 40%, (35–40) 28%, (>40) 10% | | | | C: $13 970 |
| Petrucco et al, 2007 [26] | | Institutional records | *Live birth rates* | Institutional records | 2002, 2005 | NR | *Total cost* |
| | I1: microsurgical TA in Adelaide | | I: 40% | | | | I1: AU$4850 |
| | I2: microsurgical TA in Missouri | | | | | | – |
| | C: IVF | | C: NR | | | | C: AU$6940 |
| Tan et al, 2010 [24] | | Institutional records | *Pregnancy rates / live birth rates* | Institutional records | NR | NR | *Group total cost* |
| | I1: laparoscopic TA | | I1: 77.8% / 66.7% | | | | I1: SG$85 200 |
| | I2: open TA | | I2: 70.0% / 60.0% | | | | |
| | C: IVF | | C: 46.8% / 34.6% | | | | C: SG$2 649 402 |
| Winter et al, 2012 [41] | | Literature review | NR | Literature review | 2012 | NR | NR |
| | I1: TA with more favorable prognosis | | | | | | |
| | I2: TA with less favorable prognosis | | | | | | |
| | I3: IVF | | | | | | |
| Womack et at., 2020 [42] | | Literature review | NR | Institutional records | NR | NR | NR |
| | I: TA | | | | | | |
| | C: IVF | | | | | | |

*(Continued)*

**Table 2.** (Continued)

| Study (Authors, year) | Intervention | Source of outcomes | Outcomes | Source of costs | Conversion year | Discount rate | Costs |
|---|---|---|---|---|---|---|---|
| **Vasectomy reversal** | | | | | | | |
| Cheng et al, 2021 [37] | | Literature review | *Live birth rates (by maternal age)* VR (35–37) 10.7–26.7%, (38–40) 7.8–19.5%, (>40) 4.6–11.5% IVF/ICSI (35–37) 28%, (38–40) 18.7%, (>40) 6.5% *Estimated QALY* Patient with a healthy child following fertility treatment: 0.93 Infertile patient with the desire for a child: 0.86 | Institutional records | NR | NR | *Total cost* |
| | I1: VR, then NC | | | | | | I1: $7150 |
| | I2: SR, then IVF/ICSI | | | | | | I2: $18 943 |
| | I3: back-up VR/SR, then VF/ICSI, followed by NC | | | | | | I3: $24 368 |
| | I4: back-up VR/SR, then NC, followed by IVF/ICSI | | | | | | I4: $24 368 |
| Craig et al, 2017 [36] | | Literature review | *QALY* Infertile couple: 0.56 Infertile couple who becomes pregnant: 0.63 | Institutional records | NR | NR | NR |
| | I1: VR followed by NC | | | | | | |
| | I2: TESE followed by IVF | | | | | | |
| | I3: back-up VR, then NC, followed by IVF | | | | | | |
| | I4: back-up VR, then IVF, followed by NC | | | | | | |
| Deck et al, 2000 [35] | | Institutional records | *Live birth rates* | Institutional records | NR | NR | *Total cost* |
| | I: Microsurgical VR | | I: 17% | | | | I: $4850 |
| | C: TESE and IVF/ICSI | | C: 8% | | | | C: $8315 for TESE and 1 cycle of IVF/ICSI |
| Donovan et al, 1998 [32] | | Institutional records | *Pregnancy rates / delivery rates* | Institutional records | NR | NR | *Total cost* |
| | I: Repeat VR | | I: 44% / 44% | | | | I: $9284 |
| | C: MESA and ICSI/IVF | | C: 67% / 56% | | | | C: $17 092 |
| Heidenreich et al, 2000 [28] | | Institutional records | *Pregnancy rates / live birth rates* | Institutional records | NR | NR | *Total cost* |
| | I: VVS | | I: 52% / 52% | | | | I: (VVS) €2462 |
| | C: MESA/TESE and ICSI | | C: 24.5% / live birth rates (MESA/ICSI) 22.5% (TESE/ICSI) 19.5% | | | | C: (MESA/TESE) €369, (ICSI) €2974 |

*(Continued)*

**Table 2.** (Continued)

| Study (Authors, year) | Intervention | Source of outcomes | Outcomes | Source of costs | Conversion year | Discount rate | Costs |
|---|---|---|---|---|---|---|---|
| Kassab et al, 2024 [43] | | Literature review | *Pregnancy rates by maternal age* | Literature review | NR | NR | *Total cost* |
| | I: VR | | I: (25) 76.0%, (30) 67.1%, (35) 59.0%, (40) 39.3%, (42) 11.7%, (45) 11.7% | | | | I: $12,558 |
| | C: IVF | | C: (25) 37.9%, (30) 37.9%, (35) 29.6%, (40) 20.1%, (42) 11.1%, (45) 3.5% | | | | C: $27,647 |
| Kolettis et al, 1997 [33] | | Literature review and institutional records | *Pregnancy rates / delivery rates* | Institutional records | NR | NR | *Total cost* |
| | I: VES | | I: 44% / 36% | | | | I: $8500 |
| | C: MESA and ICSI | | C: 56% / 29% | | | | C: $10 213 |
| Lee et al, 2008 [44] | | Literature review | *Pregnancy rates / delivery rate* | Literature review and institutional records | 1999, 2005 | Medical care CPI | *Procedural cost* |
| | I: VR | | I: (VR patency) 81% / 42% | | | | I: (VR) $2916 |
| | C: TESE/MESA and IVF | | C: (singleton) 70.4–72.0% (twin) 23.7–25.9% (triplet) 2.1–5.8% / 44% | | | | C: (TESE) $577, (MESA) $1439, (IVF) $12 507 |
| Meng et al, 2005 [45] | I: VR | Literature review | *Patency rates* (Bilateral VVS) 87%, (VVS/VES) 70%, (bilateral VES) 65% | Institutional records | NR | NR | *Procedural cost* (Microsurgical VVS) $10 000, (microsurgical varicocelectomy) $4500, (IVF/ICSI cycle) $10 000, (IUI cycle) $500 |
| | C: SR and ICSI | | *Pregnancy rates* (VR) 30%, (after 1 ICSI cycle) 30%, (after 4 IUI cycles) 32% | | | | |
| Pavlovich et al, 1997 [34] | | Literature review | *Pregnancy rates / delivery rates* | Literature review and institutional records | NR | NR | *Total cost* |
| | I: Microsurgical VR | | I: 52% / 47% | | | | I: (VVS) $11 922, (VES) $17 151 |
| | C: SR and ICSI | | C: NR / 33% | | | | C: (PESA/ICSI and TESE/ICSI) $20 347, (MESA/ICSI) $28 072 |
| Zhang et al, 2021 [29] | | Institutional records | *Pregnancy rates / delivery rates* | Institutional records | NR | NR | *Total cost* |
| | I: microsurgical VES | | I: 44.6% / 38.5% | | | | I: CN¥13 065 |
| | C: TESE/PESA with ICSI | | C: 27.6% / 25.8% | | | | C: CN¥40 015 |

**Abbreviations:** $, US dollar; AU$, Australian dollar; C, comparator; CN¥, Chinese yuan; €, Euro; I, intervention; ICSI, intracytoplasmic sperm injection; IVF, in vitro fertilization; MESA, microscopic epididymal sperm aspiration; NC, natural conception; NLG, Dutch guilder; NR, not reported; PESA, percutaneous epididymal sperm aspiration; QALY, quality-adjusted life-year; SG$, Singapore dollar; SR, sperm retrieval; TA, tubal anastomosis; TESE, testicular sperm extraction; VES, vasoepididymostomy; VR, vasectomy reversal; VVS, vasovasostomy.

**Table 3. Cost per outcome and study conclusions.**

| Study (Authors, year) | Intervention | Cost per outcome / ICER | Sensitivity analysis | Cost-effective? | Remark |
|---|---|---|---|---|---|
| **Tubal anastomosis** | | | | | |
| Alford et al, 2010 [38] | | *Cost per delivery (by maternal age)* | Yes, not specified | Yes | – |
| | I: Bilateral TA | I: (<35) favored by $3980 - $37 281, (41) favored by $56 686-$156 803 | | | |
| | C: IVF | C: base | | | |
| Boeckxstaens et al, 2007 [27] | | *Cost per delivery (by maternal age)* | NR | Yes | For females patients <37 years old. |
| | I: TA | I: average €6015, (<37) €4953, (≥37) €9740 | | | |
| | C: IVF | C: average €11 707, (<37) €12 140, (≥37) €11 214 | | | |
| Chua et al, 2020 [25] | | *Cost per live birth* | NR | Yes | For female patients <40 years old. |
| | I: TA | I: SG$27 109 | | | |
| | C: IVF | C: SG$52 438 | | | |
| Copperman et al, 1996 [30] | | *Cost per pregnancy* | NR | No | For female patients <40 years old. |
| | I: TA | I: $24 334 | | | |
| | C: IVF | C: $23 719 | | | |
| Haan et al, 1991 [22] | | *Cost per pregnancy* | NR | No conclusion | IVF appeared to be equivalently cost-effective as TA. |
| | I: TA | I: NLG 17 000-NLG 23 000 | | | |
| | C: Three cycles of IVF | C: NLG 22 500 | | | |
| Haan et al, 1992 [23] | | *Cost per pregnancy* | NR | No conclusion | IVF appeared to be as cost-effective as TA. |
| | I: TA | I: equivalent to IVF | | | |
| | C: IVF | C: NLG 25 000 | | | |
| Hirshfeld-Cytron et al, 2013 [39] | | *ICER: Incremental cost per live birth (by maternal age)* | One-way analyses: varying ages and underlying diseases Two-way analyses: varying IVF cost and live birth rates | Yes | TA after a prior clip or ring tubal ligation for female patients ≤40 years old. |
| | I1: TA with more favorable prognosis | I1: (≤35) base, (35–40) base, (>40) base | | | |
| | I2: TA with less favorable prognosis | I2: (≤35) dominated, (35–40) dominated, (>40 years) dominated | | | |
| | I3: IVF | I3: (≤35) dominated by $30 900, (35–40) dominated by $44 000, (>40) dominated by $9540 | | | |
| Holst et al, 1991 [31] | | *Cost per live birth* | NR | No | – |
| | I1: TA in 1980–1982 | I: $17 000 | | | |
| | I2: TA in 1986–1988 | | | | |
| | C1: All IVF in 1986–1989 | C: $12 000 | | | |
| | C2: Completed IVF treatment in 1986–1989 | | | | |

*(Continued)*

**Table 3.** (Continued)

| Study (Authors, year) | Intervention | Cost per outcome / ICER | Sensitivity analysis | Cost-effective? | Remark |
|---|---|---|---|---|---|
| Messinger et al, 2015 [40] | | *Cost per ongoing pregnancy (by maternal age)* | One-way analyses: varying IVF and TA charges | Yes | For female patients <41 years old. |
| | I: TA | I: (<35) $16 315, (35–40) $23 914, (>40) $218 742 | | | |
| | C: IVF | C: (<35) $32 814, (35–40) $45 839, (>40) $111 445 | | | |
| Petrucco et al, 2007 [26] | | *Cost per live birth (by maternal age)* | NR | Yes | For female patients ≥40 years old. |
| | I1: microsurgical TA in Adelaide | I1: AU$11 317 | | | |
| | I2: microsurgical TA in Missouri | | | | |
| | C: IVF | C: (40–42) AU$97 884 (>42) AU$182 794 | | | |
| Tan et al, 2010 [24] | | *Cost per live birth* | NR | Yes | For female patients <40 years old. |
| | I1: laparoscopic TA | I1: SG$14 200 | | | |
| | I2: open TA | | | | |
| | C: IVF | C: SG$23 446 | | | |
| Winter et al, 2012 [41] | | *ICER: Incremental cost per live birth* | Yes, not specified | Yes | For female patients ≤ 40 years old with prior ring or clip ligation. |
| | I1: TA with more favorable prognosis | I1: Dominant | | | |
| | I2: TA with less favorable prognosis | I2: Additional $11 000-$60 000 per live birth | | | |
| | I3: IVF | I3: (younger cohort) additional $30 000-$45 000 per live birth, (oldest females) additional $200 000 | | | |
| Womack et at., 2020 [42] | | NR | Yes, not specified | Yes | For female patients <40 years old. |
| | I: TA | | | | |
| | C: IVF | | | | |
| **Vasectomy reversal** | | | | | |
| Cheng et al, 2021 [37] | | *Cost per QALY (by maternal age)* | Adjusting live birth rates by maternal age | Yes | For male patients with female partners ≥35 years old. |
| | I1: VR, then NC | I1: (35–37) $7150, (38–40) $7203, (>40) $7367 | | | |
| | I2: SR, then IVF/ICSI | I2: (35–37) $40 821, (38–40) $46 247, (>40) $54 599 | | | |
| | I3: back-up VR/SR, then VF/ICSI, followed by NC | I3: (35–37) $31 289, (38–40) $33 226, (>40) $35 700 | | | |
| | I4: back-up VR/SR, then NC, followed by IVF/ICSI | I4: (35–37) $34 142, (38–40) $35 404, (>40) $37 061 | | | |

*(Continued)*

| Study (Authors, year) | Intervention | Cost per outcome / ICER | Sensitivity analysis | Cost-effective? | Remark |
|---|---|---|---|---|---|
| Craig et al, 2017 [36] | | *Cost per QALY (by maternal age)* | NR | Yes | – |
| | I1: VR followed by NC | I1: (<35) $11 349, (35–37) $11 350, (38–40) $11 485, (>40) $11 559 | | | |
| | I2: TESE followed by IVF | I2: (<35) $54 719, (35–37) $59 340, (38–40) $65 749, (>40) $38 081 | | | |
| | I3: back-up VR, then NC, followed by IVF | I3: (<35) $35 742, (35–37) $43 442, (38–40) $63 628, (>40) $41 826 | | | |
| | I4: back-up VR, then IVF, followed by NC | I4: (<35) $36 803, (35–37) $38 176, (38–40) $38 350, (>40) $38 354 | | | |
| Deck et al, 2000 [35] | | *Cost per live birth* | NR | Yes | For male patients with female partners >37 years old. |
| | I: Microsurgical VR | I: $28 530 | | | |
| | C: TESE and IVF/ICSI | C: $103 940 | | | |
| Donovan et al, 1998 [32] | | *Cost per pregnancy, cost per delivery* | Varying pregnancy and delivery rates of MESA and ICSI/IVF | Yes | – |
| | I: Repeat VR | I: $12 410, $14 892 | | | |
| | C: MESA and ICSI/IVF | C: $25 637, $35 570 | | | |
| Heidenreich et al, 2000 [28] | | *Cost per live birth* | NR | Yes | – |
| | I: VVS | I: €2793 | | | |
| | C: MESA/TESE and ICSI | C: €14 547 | | | |
| Kassab et al, 2024 [43] | | NR | NR | Yes | – |
| | I: VR | | | | |
| | C: IVF | | | | |
| Kolettis et al, 1997 [33] | | *Cost per delivery, cost per live birth* | NR | Yes | – |
| | I: VES | I: $23 611, $31 099 | | | |
| | C: MESA and ICSI | C: $35 217, $51 024 | | | |
| Lee et al, 2008 [44] | | *Cost per live birth in 1999 US dollars and 2005 US dollars* | One-way analyses: varying male procedural cost from high-volume andrology center and top quintile household income Two-way analyses: varying cost per live delivery over a range of VR patency rates and IVF delivery rates. | Yes | – |
| | I: VR | I: [1999] $19 633 \| [2005] $20 903 | | | |
| | C: TESE/MESA and IVF | C: (TESE) [1999] $45 637 \| [2005] $54 797 (MESA) [1999] $48 055 \| [2005] $56 861 | | | |
| Meng et al, 2005 [45] | | *Cost per pregnancy at 80% patency rate and 30% pregnancy rate* | One-way analyses: varying reversal patency rates at 40% constant pregnancy rate Two-way analyses: varying reversal patency and pregnancy rates | Yes | – |
| | I: VR | I: $38 983 | | | |
| | C: SR and ICSI | C: $39 506 | | | |

*(Continued)*

**Table 3.** (Continued)

| Study (Authors, year) | Intervention | Cost per outcome / ICER | Sensitivity analysis | Cost-effective? | Remark |
|---|---|---|---|---|---|
| Pavlovich et al, 1997 [34] | | *Cost per delivery* | One-way analyses: varying delivery rates | Yes | For male patients with female partner ≤39 years old. |
| | I: Microsurgical VR | I: $25 475 | | | |
| | C: SR and ICSI | C: $72 521 | | | |
| Zhang et al, 2021 [29] | | NR | NR | Yes | For male patients with female partner ≤40 years old. |
| | I: microsurgical VES | | | | |
| | C: TESE/PESA with ICSI | | | | |

**Abbreviations:** $, US dollar; AU$, Australian dollar; €, Euro; C, comparator; I, intervention; ICER, incremental cost-effectiveness ratio; ICSI, intracytoplasmic sperm injection; IVF, in vitro fertilization; MESA, microscopic epididymal sperm aspiration; NC, natural conception; NLG, Dutch guilder; NR, not reported; PESA, percutaneous epididymal sperm aspiration; QALY, quality-adjusted life-year; SG$, Singapore dollar; SR, sperm retrieval; TA, tubal anastomosis; TESE, testicular sperm extraction; VES, vasoepididymostomy; VR, vasectomy reversal; VVS, vasovasostomy.

healthcare CPI [48], Singaporean healthcare CPI [49], and United States medical care CPI [50]. We then converted values to 2024 US dollars using annual foreign exchange rates [18]. Calculation details, categorized by cost per outcome for comparison and stratified by maternal age, appear in S3 Table. Owing to substantial heterogeneity, we summarized findings in a table, as shown in Table 4, and performed no meta-analyses or sensitivity analyses.

**TA findings.** Studies conducted before 2000 favored ARTs over TA [30,31] or concluded that ARTs were as cost-effective as TA [22,23]. However, all studies conducted after 2000 concluded that TA was more cost-effective than ARTs for females younger than 40 or 41 years, with thresholds varying by study. One study reported higher cost per pregnancy for TA than for ARTs among females older than 40 years ($283 332 versus $144 352) [40].

**VR findings.** VR was more cost-effective than ARTs in all studies, regardless of female partner age. Although two studies included multiple arms evaluating a combination of VR and ARTs [36,37], only isolated VR and ART arms were used for calculation. Cost estimates varied widely; however, 1 longitudinal analysis reported a lower cost per live birth for VR in 2005 than in 1999 [44].

## Assessment of study quality

Fig 3 summarizes CHEERS 2022 compliance by evaluation category. Almost all studies met more than 20 of 28 items; however, only 8 studies met at least 23 of 28 items (> 80%) [23,27,32,34,37,39,40,44]. One study met fewer than 10 items [42]. Six studies reported the year of currency conversion [26,38–41,44], and only 3 specified a discount rate [39,40,44]. Detailed study-level assessments appear in S4 Table.

## Discussion

Cost-effectiveness patterns differed by sex. In the TA group, studies conducted before 2000 reported ARTs to be either more cost-effective or as cost-effective as TA for achieving pregnancy. These findings align with the results of the previous systematic review published in 2002 [15]. In 1 study, the total cost of ART exceeded that of TA ($22 288.04 versus $20 039.74) [30]. However, the cost per pregnancy was slightly lower with ART ($23 718.80 versus $24 333.89) owing to a higher incidence of multiple gestations in the ART group. Despite higher total costs, multiple gestations reduced the cost per pregnancy. These findings aligned with the rapid expansion of ART availability after the first birth from IVF in 1978 [51].

**Table 4. Cost per outcome of sterilization reversal versus assisted reproductive technology in 2024 US dollars.**

| Author, study year | Cost per outcome | Maternal age (years) | | | | | | | | | | Cost-effective?* | Maternal age remark |
|---|---|---|---|---|---|---|---|---|---|---|---|---|---|
| | | <35 | 35 | 36 | 37 | 38 | 39 | 40 | 41 | 42 | >42 | | |
| **Tubal anastomosis** | | | | | | | | | | | | | |
| Haan, 1991 [22] | Cost per pregnancy | NR | | | | | | | | | | Inconclusive | – |
| Holst, 1991 [31] | Cost per live birth | $54 149 / **$38 223** | | | | | | | | | | No | – |
| Haan, 1992 [23] | Cost per pregnancy | NR | | | | | | | | | | Inconclusive | – |
| Copperman, 1996 [30] | Cost per pregnancy | $60 107 / **$58 588** | | | | | NR | | | | | No | <40 |
| Petrucco, 2005 [26] | Cost per live birth | NR | | | | | **$38 020** / $392 024 | | | **$38 020** / $732 087 | | Yes | ≥40 |
| Boeckxstaens, 2007 [27] | Cost per delivery | **$4944** / $12 119 | **$9723** / $11 194 | | | | | | | | | Yes | <37 |
| Alford, 2010 [38] | Cost per delivery | NR | | | | | | | | | | Yes | – |
| Tan et al, 2010 [24] | Cost per live birth | **$14 342** / $23 680 | | | | | NR | | | | | Yes | <40 |
| Winter, 2012 [41] | Incremental cost per live birth | NR | | | | | | | | | | Yes | ≤40 |
| Hirshfeld-Cytron, 2013 [39] | Incremental cost per live birth | NR | | | | | | | | | | Yes | ≤40 |
| Messinger, 2015 [40] | Cost per pregnancy | **$21 132** / $42 503 | **$30 975** / $59 374 | | | | $283 332 / **$144 352** | | | | | Yes | <41 |
| Chua, 2020 [25] | Cost per live birth | **$22 766** / $44 037 | | | | | NR | | | | | Yes | <40 |
| Womack, 2020 [42] | Not reported | NR | | | | | | | | | | Yes | <40 |
| **Vasectomy reversal** | | | | | | | | | | | | | |
| Kolettis, 1997 [33] | Cost per delivery | **$56 749** / $74 747 | | | | | | | | | | Yes | – |
| Kolettis, 1997 [33] | Cost per live birth | **$74 747** / $122 636 | | | | | | | | | | Yes | – |
| Pavlovich, 1997 [34] | Cost per delivery | **$61 229** / $174 304 | | | | | NR | | | | | Yes | ≤39 |
| Donovan, 1998 [32] | Cost per pregnancy | **$28 899** / $34 679 | | | | | | | | | | Yes | – |
| Donovan, 1998 [32] | Cost per delivery | **$59 701** / $82 833 | | | | | | | | | | Yes | – |
| Lee, 1999 [44] | Cost per live birth | **$44 181** / (TESE) $102 699, (MESA) $108 140 | | | | | | | | | | Yes | – |
| Deck, 2000 [35] | Cost per live birth | NR | | | | **$61 693** / $224 758 | | | | | | Yes | >37 |
| Heidenreich, 2000 [28] | Cost per live birth | **$3968** / $20 669 | | | | | | | | | | Yes | – |
| Lee, 2005 [44] | Cost per live birth | **$36 464** / (TESE) $95 589, (MESA) $99 190 | | | | | | | | | | Yes | – |
| Meng, 2005 [45] | Cost per pregnancy | **$68 003** / $68 915 | | | | | | | | | | Yes | – |
| Craig, 2017 [36] | Cost per QALY | **$13 463** / $64 910 | **$13 464** / $70 392 | | **$13 624** / $77 994 | | **$13 712** / $45 173 | | | | | Yes | – |

*(Continued)*

**Table 4.** (Continued)

| Author, study year | Cost per outcome | Maternal age (years) | | | | | | | | | | Cost-effective?* | Maternal age remark |
|---|---|---|---|---|---|---|---|---|---|---|---|---|---|
| | | <35 | 35 | 36 | 37 | 38 | 39 | 40 | 41 | 42 | >42 | | |
| Cheng, 2021 [37] | Cost per QALY | NR | $7675 / $43 819 | | | $7732 / $49 644 | $7908 / $58 609 | | | | | Yes | ≥35 |
| Zhang, 2021 [29] | Not reported | NR | | | | | | | | | | Yes | ≤40 |
| Kassab, 2024 [43] | Not reported | NR | | | | | | | | | | Yes | – |

White cells indicate that sterilization reversal is more cost-effective, gray cells indicate that assisted reproductive technology is more cost-effective, while studies with black cells are not applicable for reporting economic outcome measured in cost per clinical outcome.

* As compared with assisted reproductive technologies.

**Abbreviations:** $, US dollars; NR, not reported; QALY, quality-adjusted life-year.

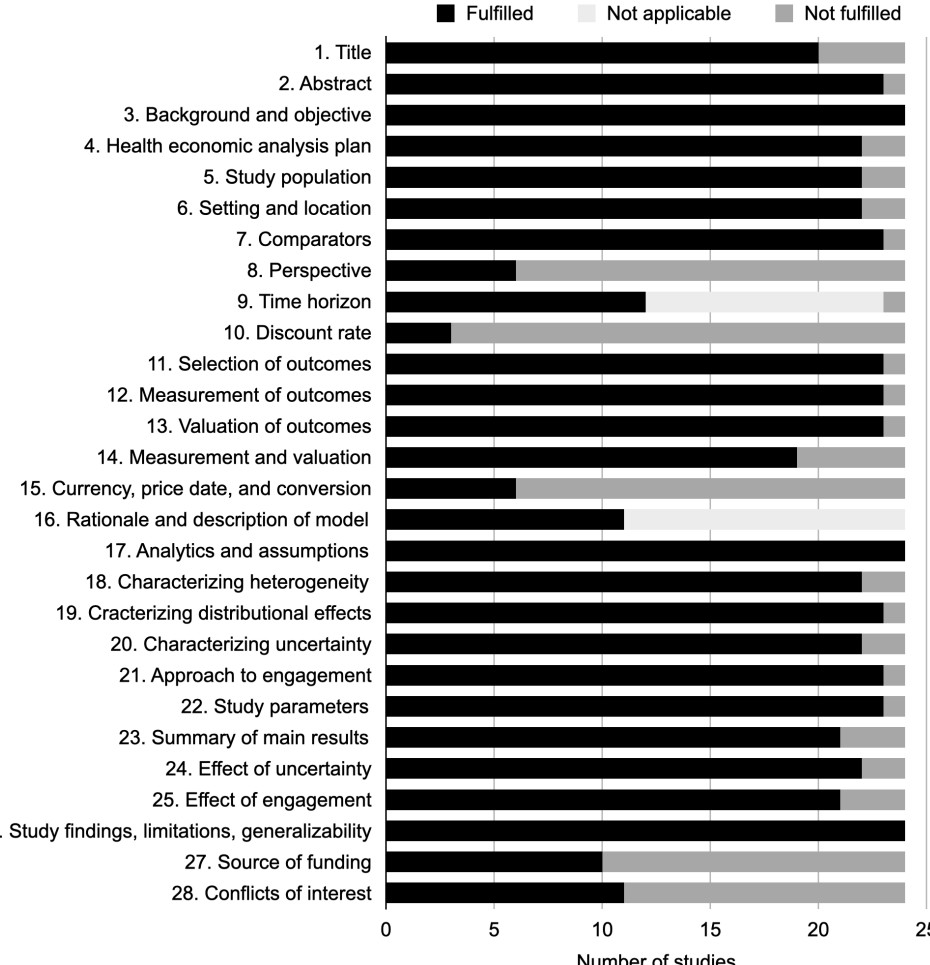

**Fig 3. Visualization of Consolidated Health Economic Evaluation Reporting Standards 2022 checklist.**

After 2000, all studies favored TA over ARTs, particularly for females younger than 40 or 41 years, reflecting the sharp fertility decline after age 40. This growing favor toward TA in this younger cohort demonstrates a longitudinal shift in economic viability, likely driven by modern advancements in microsurgical techniques. Although ART success is substantially lower in this age group than in younger cohorts [52], 1 model-based analysis concluded that ART was more cost-effective than TA ($111 445 versus $218 742 per ongoing pregnancy) [40]. These findings suggest that ARTs are more economically viable for females of advanced maternal age; therefore, females in this age group should proceed directly to ARTs. Nevertheless, due to an increasing trend of opportunistic salpingectomy for the prophylaxis of tubo-ovarian carcinoma [5], a growing cohort of female patients lacks fallopian tubes required for re-anastomosis. For these individuals, ARTs are the only options to regain fertility, regardless of age.

VR studies demonstrated consensus despite differences in reported outcomes. Consistent with TA findings, 2 studies reported lower cost per outcome in male patients with younger female partners [36,37]. Nonetheless, costs varied widely by country, setting, and study year. After converting costs per outcome to 2024 US dollars (**Table 3** and S3 Table), VR generally demonstrated lower costs per outcome. One longitudinal study comparing VR and ART in 1999 and 2005 found improved cost-effectiveness over time, with VR remaining more cost-effective [44].

Therefore, rather than dictating clinical practice, the economic thresholds identified in this study provide an evidence base for shared decision-making. If anatomical reversal is possible, the data support counseling females over 40 years old to proceed directly with ARTs to prevent delay in care. Conversely, for the younger female cohorts, tubal anastomosis yields a superior economic return by allowing multiple unassisted reproduction attempts. Vasectomy reversal is preferred over ARTs regardless of age in male patients, as a single successful surgery provides a more cost-effective reproduction than a recurrent financial burden in ART cycles.

Despite the limited number of studies, substantial heterogeneity existed in reported results. Metrics varied, including cost per pregnancy, cost per delivery, and cost per live birth, with inconsistent application across studies. Even within the same intervention, there was no consensus on which outcome should serve as the denominator. Outcome selection may have favored measures that made the evaluated intervention appear cost-effective [53]. Consequently, an intervention with higher total cost might appear more cost-effective when evaluated using a different clinical outcome. Although cost per live birth is the definitive metric for clinical success, many studies reported only intermediate outcomes, such as cost per pregnancy. This likely reflects the logistical constraints of surgical cohort studies, where tracking initial postoperative conception is more feasible than maintaining the long-term follow-up required to confirm a live birth. In addition, several studies failed to report key components of economic evaluation, such as the analytic perspective, time horizon, and discount rates. Therefore, thorough evaluation should consider total costs, the chosen outcome measure, and other reported metrics.

Moreover, none of the included studies reported a stratification of male partners' age. For studies evaluating tubal anastomosis, it was clinically sensible to factor in maternal age, as ovarian reserve predictably declines with advanced maternal age [54,55]. In contrast, although paternal age has been shown to negatively affect semen volume, motility, and morphology [56], a multivariate analysis revealed no significant difference in pregnancy rates after vasectomy reversal between males over and under 50 years old [57]. Therefore, while clinical pregnancy rates remain stable, future economic evaluations incorporating paternal age for vasectomy reversal could further address this gap.

Regarding cost adjustment, only 6 studies adjusted costs for inflation [26,38–41,44]. Of these, only 3 reported a discount rate [39,40,44]. Absent proper discounting, estimates drawn from different time periods may misstate the interventions' economic value. Our systematic review addressed this limitation by standardizing all costs to 2024 US dollars, enabling direct comparisons and revealing trends across 4 decades.

Another limitation observed among primary economic evaluations is the geographic concentration of evidence. Nearly all studies were conducted in high-income countries, except for 1 study from China [29], which is the sole representative of upper-middle–income countries [58]. This distribution likely reflects disparities in access to reproductive technologies

across socioeconomic settings, limiting generalizability to lower-income settings. Therefore, direct extrapolation of these findings to low- and middle-income countries should be done with considerable caution, as procedures may be structured differently. For instance, the relative costs of specialized surgical labor versus laboratory-intensive ART cycles are likely to follow different economic paradigms in resource-constraint settings. Additionally, an individuals' true health financial burden is heavily influenced by income, local health policies, and insurance coverage. These are variables that cannot be retrospectively adjusted for in this analysis.

Furthermore, most included studies did not incorporate quality-adjusted life years into their analyses. Authors may have chosen outcome-specific metrics to allow readers to assess different dimensions of effectiveness without utility weighting [53]. However, the absence of quality-adjusted life years hindered aggregation of parental, fetal, and neonatal outcomes into a single metric and shifted focus to intermediate fetal outcomes rather than final neonatal health outcomes.

## Limitations

This systematic review focused exclusively on studies directly comparing sterilization reversal with ARTs, potentially excluding relevant cost data from other clinical or surgical contexts that could inform health economic decisions. Furthermore, given the scarcity of cost adjustments and the absence of paternal age stratification of included studies, future primary economic evaluations addressing these specific variables would provide a more comprehensive understanding of sterilization reversal costs. Moreover, although we performed an extensive citation search without language restrictions, some studies, especially those published in local languages, may have been overlooked because only English search terms were used.

Nevertheless, this review represents the most current and comprehensive synthesis of evidence on sterilization reversal; the last systematic review on this topic was published in 2002 [15]. All costs were standardized to 2024 US dollars, ensuring consistency and comparability across studies. The review also encompasses both male (VR) and female (TA) sterilization reversal. These strengths position this review as the most thorough and up-to-date resource available.

## Conclusions

Assuming anatomical feasibility, TA is cost-effective for restoring fertility, particularly for females younger than 40 years. For older females, ARTs are more cost-effective. Furthermore, for patients lacking fallopian tubes due to prior salpingectomy, ARTs remain the sole viable option, regardless of age. For males with prior vasectomy, VR is more cost-effective regardless of the female partner's age. Therefore, sterilization reversal may confer an economic advantage over ARTs. However, owing to substantial methodological heterogeneity and the large proportion of evidence from high-income countries, application of these findings requires careful consideration of local infrastructure, funding mechanisms, and health policy.

## Supporting information

**S1 Table. PRISMA 2020 checklist.**
(PDF)

**S2 Table. Search terms.**
(PDF)

**S3 Table. Cost per outcome of sterilization reversal in 2024 US dollars.** * Health CPI of base year only. Only studies that reported costs per outcome were calculated. Abbreviations: ART, assisted reproductive technology; AUD, Australian dollar; CPI, consumer price index; EUR, Euro; LCU, local currency unit; NA, not applicable; QALY, quality-adjusted life-year; SGD, Singapore dollar; SR, sterilization reversal; USD, US dollar.
(PDF)

**S4 Table. Consolidated health economic evaluation reporting standards 2022 checklist.** Abbreviations: N, "not fulfilled"; NA, "not applicable"; Y, "fulfilled.".
(PDF)

## Acknowledgments

We thank Euarat Mepramoon for project coordination. We also thank David Park for English-language editing. Artificial intelligence was not utilized for literature searching and screening, data collection, analysis, or critical appraisal of the included literature. During the manuscript preparation, the authors used Gemini (Google) to improve language readability and consolidate structural ideas. After using this tool, the authors and a native-English-speaking medical-manuscript editor reviewed and edited the manuscript as needed. The authors take full responsibility for the final content of the publication.

## Author contributions

**Conceptualization:** Brandon Chongthanadon, Suvijak Untaaveesup, Chayanis Kositamongkol, Pochamana Phisalprapa, Krasean Panyakhamlerd, Vitaya Titapant.

**Data curation:** Brandon Chongthanadon, Suvijak Untaaveesup, Chayanis Kositamongkol, Pochamana Phisalprapa.

**Formal analysis:** Brandon Chongthanadon, Suvijak Untaaveesup, Chayanis Kositamongkol.

**Funding acquisition:** Pochamana Phisalprapa, Vitaya Titapant.

**Investigation:** Brandon Chongthanadon, Suvijak Untaaveesup, Chayanis Kositamongkol.

**Methodology:** Brandon Chongthanadon, Suvijak Untaaveesup, Chayanis Kositamongkol.

**Project administration:** Chayanis Kositamongkol, Pochamana Phisalprapa.

**Resources:** Chayanis Kositamongkol.

**Supervision:** Chayanis Kositamongkol, Pochamana Phisalprapa, Krasean Panyakhamlerd, Vitaya Titapant.

**Validation:** Brandon Chongthanadon, Suvijak Untaaveesup, Chayanis Kositamongkol, Pochamana Phisalprapa, Krasean Panyakhamlerd, Vitaya Titapant.

**Visualization:** Brandon Chongthanadon, Suvijak Untaaveesup, Chayanis Kositamongkol.

**Writing – original draft:** Brandon Chongthanadon, Suvijak Untaaveesup, Chayanis Kositamongkol.

**Writing – review & editing:** Brandon Chongthanadon, Suvijak Untaaveesup, Chayanis Kositamongkol, Pochamana Phisalprapa, Krasean Panyakhamlerd, Vitaya Titapant.

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
