## [Decision Letter · Decision Letter 0]

29 Mar 2026

PONE-D-25-67814Economic evaluation of sterilization reversal in infertility treatment: A systematic reviewPLOS One

Dear Dr. Kositamongkol,

Thank you for submitting your manuscript to PLOS ONE. After careful consideration, we feel that it has merit but does not fully meet PLOS ONE’s publication criteria as it currently stands. Therefore, we invite you to submit a revised version of the manuscript that addresses the points raised during the review process.

We look forward to receiving your revised manuscript.

Kind regards,

Iwaho Kikuchi, Ph. D., M.D.

Academic Editor

PLOS One

Additional Editor Comments:

Thank you for your submission and for the thoughtful work presented in this manuscript. The paper has now been evaluated by five reviewers. One reviewer recommends acceptance, three recommend minor revisions, and one recommends major revisions.

After considering all reviewer comments and the overall quality and clarity of the manuscript, I believe that the concerns raised can be addressed with minor revisions. Therefore, I am inviting you to submit a revised version of the manuscript that carefully responds to all reviewer comments.

Please ensure that:

Each point raised by the reviewers is addressed in a point‑by‑point response document.

All clarifications, methodological details, and textual revisions are incorporated into the manuscript itself.

Any changes made in response to the reviewer who requested major revisions are clearly explained, especially where you believe only partial modification is appropriate.

Once the revised manuscript and response document are submitted, I will proceed with the next stage of the editorial assessment. At this time, no further external review is anticipated, provided that the revisions adequately address the reviewers’ concerns.

Thank you again for your contribution, and I look forward to receiving your revised manuscript.

Best regards,

Editor

Reviewers' comments:

Reviewer's Responses to Questions

**Comments to the Author**

1. Is the manuscript technically sound, and do the data support the conclusions?

Reviewer #1: Partly

Reviewer #2: Yes

Reviewer #3: Yes

Reviewer #4: Yes

Reviewer #5: Yes

2. Has the statistical analysis been performed appropriately and rigorously? 

Reviewer #1: N/A

Reviewer #2: Yes

Reviewer #3: Yes

Reviewer #4: Yes

Reviewer #5: N/A

3. Have the authors made all data underlying the findings in their manuscript fully available?

Reviewer #1: Yes

Reviewer #2: Yes

Reviewer #3: Yes

Reviewer #4: Yes

Reviewer #5: Yes

4. Is the manuscript presented in an intelligible fashion and written in standard English?

Reviewer #1: Yes

Reviewer #2: Yes

Reviewer #3: Yes

Reviewer #4: Yes

Reviewer #5: Yes

5. Review Comments to the Author

Reviewer #1: “We excluded studies of same-sex couples and nonhuman subjects” - I find this sentence insensitive. Furthermore, I see no comprehensible reason why same-sex couples should be excluded.

“We categorized included studies into 2 groups: male and female.” - If a distinction was made based on binary gender, I would also refer to it as such.

I find that the study selection section lacks explanations as to why certain studies were excluded. This is essential for reflecting on which studies may be missing in the discussion. This makes the research process less transparent.

Adjustment methods; It is not only the cost of the procedure that is decisive. Due to the different situations of individuals in different healthcare systems, there are other factors that are decisive for the comparisons. For example, average income, the extent to which costs are covered by health insurance companies or similar institutions, and so on.

Discussion

The consequences of the results are not addressed sufficiently. What does it mean that IVF is made more economically accessible for older women? What does it mean that VR is cheaper for men with younger women?

There is also a lack of limitation in that male age was not surveyed, and a lack of reflection on why this was probably not done.

There is a lack of classification of what knowledge could be generated and what is missing, due to the study designs of the included studies.

Reviewer #2: 1. Introduction: Relevance and Methodological Rigor

This systematic review addresses a significant gap in reproductive medicine, as the last comprehensive economic synthesis on this topic dates back to 2002. The manuscript demonstrates commendable methodological rigor by adhering to the PRISMA 2020 guidelines and utilizing the CHEERS 2022 checklist to assess the quality of economic evaluations.

A major technical strength is the standardization of historical costs across four decades into 2024 United States dollars (USD) using local Healthcare Consumer Price Indices (CPI) . This allows for a more honest temporal comparison of the economic value of surgical reversal versus assisted reproductive technologies (ART).

2. Evaluation of Sections

Title and Abstract

• The title is clear and descriptive of the study's scope.

• The abstract follows the journal's structure, clearly outlining the objectives, search strategy (MEDLINE, Embase, Scopus), and the primary conclusion that reversal is generally more cost-effective for younger patients.

Methodology

• The use of a registered PROSPERO protocol (CRD420250588564) ensures transparency.

• Statistical Note: The authors correctly avoided a meta-analysis due to the extreme heterogeneity in study designs and reported metrics.

Quality of References

The bibliography includes high-impact journals such as Human Reproduction Update and Fertility and Sterility. However, there are notable concerns:

• Geographic Bias: 16 out of the 24 included studies originate from the United States. This limits the global applicability of the findings.

• Economic Quality: The authors acknowledge that only 8 of 24 studies met more than 80% of the CHEERS criteria, indicating that the economic evidence base is of variable quality.

• Technological Lag: While the clinical references for ART are appropriate, the surgical references lack discussion on modern prophylactic paradigms.

3. Critical Analysis and Major Concerns

A. Generalizability and LMIC Context

The findings are heavily skewed toward high-income country data . The authors must add a paragraph cautioning against the direct extrapolation of these costs to Low- and Middle-Income Countries (LMICs). In these regions, the valuation of health and the cost of specialized microsurgical labor versus laboratory-intensive ART cycles may follow entirely different economic structures.

B. Outcome Inconsistency: Pregnancy vs. Live Birth

There is a lack of standardization between "cost per pregnancy" and "cost per live birth" . For clinical counseling, the live birth rate is the only definitive metric for success. The authors should side-by-side these rates more explicitly to clarify if the economic advantage of surgery persists when only live births are considered.

C. The Paradigm of Opportunistic Salpingectomy (Major Omission)

The most critical clinical flaw is the failure to discuss opportunistic salpingectomy. Current international guidelines, including the 2026 ESGO consensus published in JAMA, strongly advocate for total salpingectomy over traditional tubal ligation (salpingotripsy) for ovarian cancer prophylaxis.

• If total salpingectomy is performed as the standard for permanent contraception, tubal anastomosis becomes anatomically impossible.

• The authors must discuss how this shift toward non-reversible, cancer-preventive surgery impacts the future relevance of their economic model for female patients.

4. Final Recommendation

The manuscript is technically well-written and serves as a valuable historical and economic update. However, to be suitable for a high-impact journal like PLOS ONE, it must bridge the gap between economic theory and modern surgical-oncological reality.

Recommendation: Major Revision

The authors must:

1. Incorporate the ESGO/JAMA 2026 paradigm on opportunistic salpingectomy.

2. Explicitly state the limitations of generalizing US-centric data to diverse global health systems.

3. Clarify the clinical weight of live birth rates versus intermediate pregnancy outcomes.

Reviewer #3: What was the date of inception of the data search? I.e. date of earliest data from MEDLINE, etc. Also, number of patients is small. Understand the limitations of the search but would benefit from more numbers to better help determine the cost-effectiveness. Also, would be good to see if there is more data in less well-developed areas to help validate the findings.

Reviewer #4: The manuscript presents a systematic review on the economic evaluation of sterilization reversal in infertility treatment. The topic is relevant for reproductive health decision-making with regard to fertility. The manuscript is generally well structured and has all the relevant supporting documents uploaded; however, a few reporting issues require clarification.

1. Clarification of intervention groups

Two of the included studies, Cheng et al 2021 (29) and Craig et al, 2017 (28), have the intervention group receiving sterilization (VR) followed by an artificial reproductive technique. This introduces potential confounding because the intervention group is exposed to two procedures. The authors should consider discussing its potential impact on the overall findings and conclusions

2. Unclear statement

The statement “However, time horizons marked as ‘not applicable’ for model analyses, which focus on evaluating at a single point in time” appears incomplete. The authors may wish to revise this sentence.

Reviewer #5: This is a review of the manuscript titled “Economic evaluation of sterilization reversal in infertility treatment: A systematic review”

My overall impression is that this is a well-conducted and well-written systematic review. This is an interesting contribution to the literature, however, it is suggested that a rationale of the inclusion of older studies and summary of their results from the previous review be made due to the known results from the prior systematic review. This SR could be reframed as an "update" and contribution to the literature.

This manuscript could be improved with the following recommendations:

1. Line 46: consider using either effectiveness for both vasectomy and tubal ligation, or pregnancy rate for both instead of comparing effectiveness with pregnancy rate to the reader.

2. Lines 57-63: good explanation of the reversals, consider adding complications such as ectopic pregnancy with TA, other pregnancy losses, etc.

3. Consider including the research question and hypothesis in the Introduction or Methods section to show why this study is unique and important, when the authors point out that a prior systematic review has been conducted in 2002.

4. Lines 53-56: ART and IVF and other procedure definitions are included, which is important, however, it seems these terms are used interchangeably throughout the manuscript based on the study. If ART includes IVF, consider using that language throughout to avoid confusion, as this is the term used in the abstract and conclusion.

5. Line 90: states “irrespective of language” Please clarify how the authors read/ comprehended publications not among those that they had literacy in.

6. Line 134: excellent use of the CHEERS checklist for this study

7. Line 154: consider mentioning that the only study included, conducted in a LMIC, was that of China

8. Line 234: the TA studies compared cost effectiveness to IVF and ART’s however the VR studies compared only cost effectiveness to ART’s, consider mentioning that comparison to IVF specifically was missing in the studies included or use ART as the overarching language throughout as mentioned previously.

9. Line 296: consider adding a sub-section starting here : “Limitations”

10. Line 296 : Cost per pregnancy: consider mentioning the possible outcomes: SAB, ectopic, preterm, live birth, etc., either here or in the Introduction, for the reader to be aware of the increased risk of ectopic pregnancy with TA.

11. Table 2 and Discussion section lines 326-27: Ten (10) out of the 24 studies included were from prior to 2002, which was when, as the authors point out, the last systematic review of the economic costs of sterilization reversal was conducted. Five (5) of these studies already reviewed, were also included in this review (Copperman, 1996; Deck, 2000; Donovan, 1998; Kolettis, 1997; and Palovich, 1997). Consider giving rationale as to why were they included here. Consider also including years of the included studies (x before 2002 and Y since the prior systematic review) in the “Results” section. Consider mentioning in the Limitations section, a comparison to prior findings from the 2002 article. Since the conclusions of the 2002 systematic review article according to the authors was that IVF/ ART was more cost effective (similar to your own older <2002 TA studies included here) how does including these articles in this review, contribute to the literature? Consider analyzing older studies separately.

a. Garceau L, Henderson J, Davis LJ, Petrou S, Henderson LR, McVeigh E, et al. Economic implications of assisted reproductive techniques: a systematic review. Hum Reprod. 2002;17(12):3090-109.

12. Consider adding a statement at the conclusion of the manuscript stating whether AI was used in any of the analysis, writing, editing etc.

6. PLOS authors have the option to publish the peer review history of their article (what does this mean?). If published, this will include your full peer review and any attached files.

Reviewer #1: **Yes:** Lisa Glaum

Reviewer #2: No

Reviewer #3: No

Reviewer #4: No

Reviewer #5: No

---

## [Author Response · Author response to Decision Letter 1]

23 Apr 2026

Dear Editor,

On behalf of all authors, I am pleased to submit our revised manuscript, “Economic evaluation of sterilization reversal in infertility treatment: A systematic review,” for consideration for publication in PLOS ONE.

We carefully addressed all comments and suggestions from the editor and reviewers. The manuscript has been thoroughly revised. In the accompanying point-by-point response, we provide detailed answers to each issue raised. Please see the attached files in the system. We are grateful for the constructive feedback, which has improved the quality, transparency, and clarity of our work.

Sincerely,

Chayanis Kositamongkol, PharmD, MSc

Corresponding Author

Division of Ambulatory Medicine, Department of Medicine

Faculty of Medicine Siriraj Hospital, Mahidol University

2 Wang Lang Road, Bangkok Noi

Bangkok 10700, Thailand

chayanis.kos@mahidol.ac.th

---

## [Editor Report · Decision Letter 1]

12 May 2026

Economic evaluation of sterilization reversal in infertility treatment: A systematic review

PONE-D-25-67814R1

Dear Dr. Kositamongkol,

We’re pleased to inform you that your manuscript has been judged scientifically suitable for publication and will be formally accepted for publication once it meets all outstanding technical requirements.

Kind regards,

Iwaho Kikuchi, Ph. D., M.D.

Academic Editor

PLOS One

Additional Editor Comments (optional):

Thank you for your thorough and thoughtful revisions. I have reviewed the revised manuscript as well as your detailed responses to the reviewers’ comments.

All concerns raised by the reviewers have been adequately addressed. The methodological clarifications, updated literature search, and improved transparency in the economic analysis have strengthened the manuscript. No further revisions are required.

I am pleased to inform you that your manuscript is accepted for publication in PLOS ONE.

Congratulations, and thank you for your contribution.
---

## [Editor Report · Acceptance letter]

PONE-D-25-67814R1

PLOS One

Dear Dr. Kositamongkol,

I'm pleased to inform you that your manuscript has been deemed suitable for publication in PLOS One. Congratulations! Your manuscript is now being handed over to our production team.

Kind regards,

on behalf of

Dr. Iwaho Kikuchi

Academic Editor

PLOS One